# FreeEyeglass: Training-free and Target-mask-free Eyeglass Transfer for Facial Videos

## Abstract

The rise of e-commerce and short-video platforms has fueled demand for realistic video-based virtual try-on. Unlike virtual try-on of clothing, which has been actively studied so far, virtual try-on of eyeglasses is uniquely challenging: they physically interact with facial geometry, and they strongly affect facial identity, making faithful preservation of unedited regions especially important. Existing generative editing approaches, such as GAN- and diffusion-based methods, lack reconstruction objectives and often rely on inpainting, which fails to ensure identity consistency. We argue that semantic editing requires not only plausible generation but also faithful reconstruction, making autoencoder-based latent spaces particularly suitable. We introduce a training-free, reference-guided framework for video eyeglass transfer built on Diffusion Autoencoders (DiffAE). By blending semantic features in the encoder and incorporating spatial-temporal self-attention, our method achieves realistic, identity-preserving, and temporally consistent results, and points to the potential of autoencoder-based latent spaces for local video editing. Our implementations and datasets will be released upon acceptance.

## 1 Introduction

The rapid growth of e-commerce and short-video platforms has created a strong demand for realistic video-based virtual try-on systems. While most existing research has centered on clothing in the video setting (Jiang et al., 2022; Xu et al., 2024; Fang et al., 2024; Wang et al., 2024b; Nguyen et al., 2025), work on other wearable objects has been explored only in the image domain (Miao et al., 2025; Feng et al., 2025). Among them, eyeglasses stand out as a particularly important category. They have long been treated as a standalone research topic in vision and graphics, with prior work on try-on (Zhang et al., 2017; Li et al., 2023), detection (Wu et al., 2002; Bekhet & Alahmer, 2021), removal (Lyu et al., 2022; Zhang & Guo, 2025), and product design (Bai et al., 2021; Plesh et al., 2023). Compared to clothing and other accessories, eyeglasses pose uniquely complex challenges. Geometrically, they must fit precisely on the nose and ears rather than being simply overlaid on the face. They also overlap directly with the eyes and eyebrows, regions known to be critical for facial identity perception (Schyns et al., 2002; Tanaka & Simonyi, 2016). Realistic video eyeglass transfer, therefore, requires not only rendering the glasses plausibly but also preserving identity and maintaining temporal coherence under motion and viewpoint changes. Despite this practical importance, the problem has not been systematically studied to date.

Despite rapid advances in semantic image and video editing, the generative models that are widely adopted, GANs (Karras et al., 2019; 2020b;a) and text-conditional diffusion models (Rombach et al., 2022), are not inherently well-suited for local editing tasks such as object transfer. Trained primarily for generation, these models typically lack reconstruction ability, which makes them prone to alter unedited regions when adapted to editing. To enforce more localized changes, many object transfer methods (Yang et al., 2023a; Chen et al., 2024b;a; Song et al., 2024; Jiang et al., 2025) train or finetune these models with an inpainting objective, where masked regions are filled with reference content. While this improves locality, they are still unable to handle the unique challenges of eyeglass transfer. Masks inevitably discard identity-related content and spatial information around the eyes, leading to artifacts, poor harmonization, and identity inconsistency. These issues are further amplified in videos, where temporal coherence must also be maintained, and the results often resemble simply copying the reference into the target region.

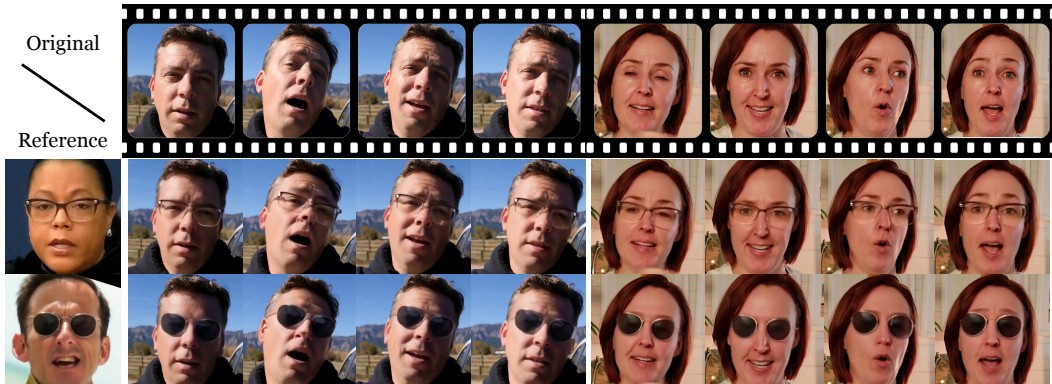

Figure 1: **FreeEyeglass** is a training-free reference-based video editing method for transferring reference eyeglasses to target facial videos semantically while achieving harmonious local editing and temporal consistency. Given a reference facial image specified with eyeglass position, our method does not require the frame-by-frame accurate masks on the target videos for transferring eyeglasses, which has been required in state-of-the-art inpainting-based editing methods.

These shortcomings indicate that eyeglass transfer requires more than inpainting-based generation. What is needed is an approach that can edit locally while preserving critical identity information. Generative models that lack reconstruction objectives struggle to provide this guarantee. Autoencoders, in contrast, are trained with explicit reconstruction objectives, equipping them with a strong capability to retain unedited content. Diffusion autoencoders (DiffAE) (Preechakul et al., 2022) further show that their latent spaces are compact and semantically structured, enabling natural local edits with minimal distortion. Through extensive experimentation, we observe that DiffAE's semantic latent space provides not only appearance preservation but also geometric and positional alignment. Even with a coarse, face-aligned reference mask without any explicit geometric modeling, DiffAE naturally adapts the reference eyeglasses to the target face, correcting their placement and visible geometry across frames. This suggests that the DiffAE's semantic latent space, despite not being trained for any generative editing objectives may in fact be better suited for eyeglass transfer in video, where both identity preservation and local realism are critical.

Building on this insight, we leverage DiffAE as a backbone and introduce a training-free, target-mask-free framework for reference-based eyeglass transfer in video, as shown in Fig. 1. While DiffAE has been explored with text or classifier guidance (Kim et al., 2023), the approach to achieving reference-based editing with this model has not been studied. We address this by designing a feature-blending mechanism in the semantic encoder, where reference and target features are combined to construct new semantic features for each frame. These are used in DDIM inversion with noise blending to guide the rendering of eyeglasses on the target face. Furthermore, we extend this framework to video by incorporating spatial-temporal self-attention focused on the editing region to enhance temporal consistency. Our method achieves realistic eyeglass transfer that preserves identity and adapts to pose changes. It demonstrates the potential of compact autoencoder latents for local video editing.

Our contributions are summarized as follows:

- We present the first *training-free and target-mask-free* framework for **reference-based video eyeglass transfer** to tackle a practically important yet underexplored problem in virtual try-on.
- We show how to adapt diffusion autoencoders for reference-based editing by designing a simple feature blending strategy in the semantic encoder, combined with a spatial-temporal self-attention to ensure natural placement and temporal consistency of eyeglasses.
- We establish a comprehensive benchmark for the video eyeglass transfer task, which will be publicly released (except for the CG face dataset).

## 2 RELATED WORK

**Eyeglasses** Eyeglasses are a distinctive accessory that strongly influences facial perception and identity, and have therefore become a standalone research topic in computer vision and graphics. Prior works span several directions, including virtual try-on (Li & Yang, 2011; Yuan et al., 2011; Niswar et al., 2011; Huang et al., 2012; 2013; Tang et al., 2014; Zhang et al., 2017; Li et al., 2023), eyeglass detection (Wu et al., 2002; Bekhet & Alahmer, 2021), removal (Hu et al., 2020; Lee &

Lai, 2020; Lyu et al., 2022; Zhang & Guo, 2025; Arkushin et al., 2025), and customized product design (Bai et al., 2021; Plesh et al., 2023). In the context of try-on of eyeglasses, most methods adopt predefined 3D eyeglass models and composite them onto faces (Li & Yang, 2011; Yuan et al., 2011; Niswar et al., 2011; Huang et al., 2012; 2013; Tang et al., 2014). Recent techniques simulate advanced physical effects like refraction (Zhang et al., 2017) or utilize multi-view data for realistic avatar reconstruction (Li et al., 2023). Though effective, they constrain users to predefined shapes and styles. Compared to these prior works, our work addresses the problem of adding and transferring arbitrary reference eyeglasses onto facial videos from a single reference image of eyeglasses without relying on predefined 3D models or extensive multi-view datasets.

**Facial video editing**  Facial video editing has been explored through latent-space manipulation (Shen et al., 2020; Yao et al., 2021; Patashnik et al., 2021) and temporal consistency techniques such as smoothing and optical flow (Tzaban et al., 2022; Alaluf et al., 2022; Xu et al., 2022), often built on StyleGAN (Karras et al., 2019; 2020b). More recent models, including StyleGANEX (Yang et al., 2023b) and FED-NeRF (Zhang et al., 2024), reduce reliance on cropping and alignment. Kim et al. (2023) introduces a DiffAE-based method that significantly improves reconstruction quality but remains limited to classifier- or text-guided editing. S3Editor (Wang et al., 2024a) offers a model-agnostic self-training framework for semantic disentanglement but similarly lacks fine-grained control. Our work adopts DiffAE (Preechakul et al., 2022) to ensure reconstruction fidelity while enabling reference-guided semantic editing for facial videos.

**Inpainting-based image and video editing**  Inpainting-based editing is a common strategy for object insertion, where masked regions are filled with plausible content guided by a reference. Recent diffusion-backboned models (Song et al., 2023; Yang et al., 2023a; Chen et al., 2024b; Song et al., 2024; Chen et al., 2024a) achieve good semantic coherence, but when applied frame by frame, they struggle with temporal consistency. Training-free variants such as TF-ICON (Lu et al., 2023) reduce the need for fine-tuning but still require accurate masks. Video-oriented extensions, including VideoAnyDoor (Tu et al., 2025) and VACE (Jiang et al., 2025), improve temporal alignment but remain mask-dependent. This mask-based formulation is reasonable for generic object insertion, but it is ill-suited for eyeglass transfer. As inpainting restricts editing to masked regions, these methods treat the eyeglass area independently of the surrounding face, which often weakens harmonization with facial geometry and identity. Our approach differs by integrating reference features directly into the semantic encoder, allowing the model to adapt eyeglasses to pose and context without relying on explicit target masks, which leads to more natural and temporally consistent results in video.

## 3 METHOD: FREEEYEGLASS

Figure 2 illustrates the overview of our FreeEyeglass pipeline. In this section, we first recap the Diffusion Autoencoder (DiffAE) (Preechakul et al., 2022) and explain how we semantically place the eyeglasses with a pretrained DiffAE.

### 3.1 PRELIMINARY: DIFFUSION AUTOENCODER (DIFFAE)

Our method is built on DiffAE proposed in Preechakul et al. (2022). In DiffAE, given an input image $\mathbf{x}$, the semantic encoder $\mathrm{Enc}(\mathbf{x})$ maps it to a semantically meaningful latent $\mathbf{z}_{\mathrm{sem}}$. A stochastic latent $\mathbf{z}_{\mathrm{sto}}$, which captures the remaining stochastic details to achieve a near-perfect reconstruction, is then computed through the deterministic DDIM inversion (Song et al., 2021) using a conditional diffusion model $\mathrm{DDIM}(\mathbf{z}_{\mathrm{sto}}, \mathbf{z}_{\mathrm{sem}})$. Taking $\mathbf{z}_{\mathrm{sto}}$ and $\mathbf{z}_{\mathrm{sem}}$ as input, the conditional diffusion model $\mathrm{DDIM}(\mathbf{z}_{\mathrm{sto}}, \mathbf{z}_{\mathrm{sem}})$ reconstructs an image $\hat{\mathbf{x}}$ through the generative DDIM process. A typical DiffAE model uses a U-Net architecture (Ronneberger et al., 2015) similar to the diffusion model proposed in Dhariwal & Nichol (2021), which consists of multiple ResBlocks (He et al., 2016) and self-attention blocks (Vaswani et al., 2017). The semantic encoder shares the same architecture as the U-Net encoder and conditions the diffusion U-Net by adaptive group normalization (Dhariwal & Nichol, 2021; Huang & Belongie, 2017).

### 3.2 FEATURE BLENDING FOR EYEGLASS TRANSFER

The semantic encoder $\mathrm{Enc}(\cdot)$ is designed to encode the semantic information of the input image. In our case, the input images are the aligned faces from video frames. The semantic encoder maps these input images to a latent space where high-level semantic features are captured. Our approach involves blending the feature maps of the target frames with those of the reference eyeglasses at each ResBlock of the semantic encoder. By doing so, we aim to incorporate the semantics of the reference

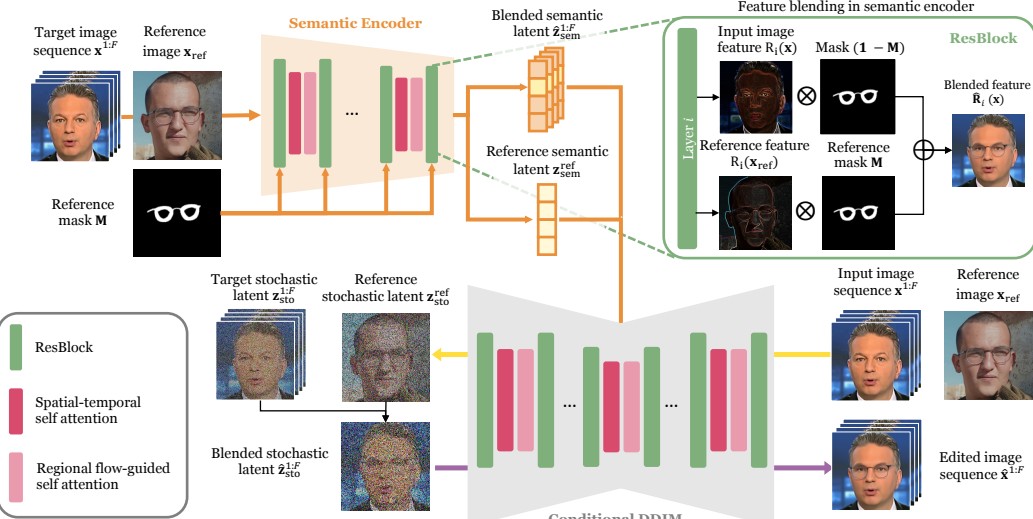

Figure 2: **Overview of our FreeEyeglass pipeline.** Given a target video (*i.e.*, target image sequence) and a reference image of desired eyeglasses, we blend the features of the reference eyeglasses and the target image sequence to obtain a blended semantic latent sequence. We then compute the stochastic latent sequences with the input images and semantic latent sequences through conditional DDIM inversion and construct a blended stochastic latent sequence. Using the blended stochastic latent and semantic latent sequences, we can obtain the final edited image sequence with our desired eyeglasses semantically and naturally placed through condition DDIM sampling.

eyeglasses into the semantic latent of the target frame. The blending of feature maps is performed using a binary mask at each ResBlock. We represent each ResBlock $i$ as a function $\mathbf{R}_i(\mathbf{x})$. Let a target frame be $\mathbf{x}$, a reference image be $\mathbf{x}_{\mathrm{ref}}$, and let $\mathbf{M} \in \{0,1\}^{h_i \times w_i}$ be the binary mask indicating the eyeglass region. Here, $h_i$ and $w_i$ are the spatial dimensions of the feature map $\mathbf{R}_i(\cdot)$, and the mask is bilinearly downsampled to match this resolution. We compute the blended feature map $\hat{\mathbf{R}}_i(\mathbf{x})$ as follows:

$$\hat{\mathbf{R}}_i(\mathbf{x}) = \mathbf{M} \odot \mathbf{R}_i(\mathbf{x}_{\mathrm{ref}}) + (\mathbf{1} - \mathbf{M}) \odot \mathbf{R}_i(\mathbf{x}), \tag{1}$$

where $\odot$ denotes element-wise multiplication. The mask $\mathbf{M}$ ensures that the features from the reference eyeglasses are only blended into the specified regions of the target frame.

After the feature blending process, we obtain a new semantic latent $\hat{\mathbf{z}}_{\mathrm{sem}}$ that merges the semantic information from the target frame $\mathbf{x}$ and the reference image $\mathbf{x}$ref. While integrating the reference eyeglasses into the semantic latent allows the model to place eyeglasses onto the target image semantically, this operation can introduce local noise and boundary artifacts due to feature mismatches between the target and reference features. To mitigate these artifacts, we also blend the stochastic latent codes, which capture residual information from the input image $\mathbf{x}$ that is not represented in the semantic latent $\hat{\mathbf{z}}_{\mathrm{sem}}$. By mixing the stochastic latent of the reference image latent $\mathbf{z}_{\mathrm{sto}}^{\mathrm{ref}}$ with that of the target image $\mathbf{z}_{\mathrm{sto}}$, we smooth the transition between reference and target features and suppress boundary inconsistencies. We construct a stochastic-latent blending mask $\tilde{\mathbf{M}}$ by combining the original reference mask $\mathbf{M}$ and its Gaussian-blurred version $\mathrm{Blur}(\mathbf{M})$:

$$\tilde{\mathbf{M}} = \beta \mathbf{M} + \gamma \, \mathrm{Blur}(\mathbf{M}), \tag{2}$$

where $\beta \geq 0$ and $\gamma \geq 0$ are scalar parameters that control the intensity and smoothness of the blending between the stochastic latent codes. We clamp the $\tilde{\mathbf{M}}$ values to ensure they lie within the $[0, 1]$ range. We smooth only the border of the blending mask, keeping the interior binary so that high-frequency details are preserved. Using the constructed mask $\tilde{\mathbf{M}}$, we perform alpha blending of the stochastic latent codes of the target and reference images:

$$\hat{\mathbf{x}}_T = (1 - \tilde{\mathbf{M}}) \odot \mathbf{z}_{\mathrm{sto}} + \tilde{\mathbf{M}} \odot \mathbf{z}_{\mathrm{sto}}^{\mathrm{ref}}, \tag{3}$$

where $\odot$ denotes element-wise multiplication. Using the blended stochastic latent $\hat{\mathbf{z}}_{\mathrm{sto}}$ as the starting noise of the DDIM sampling, we ensure that the generated image not only semantically includes the eyeglasses but also preserves their specific shape and color, significantly improving the preservation of fine details in the edited images.

## 3.3 SELF-ATTENTION FOR TEMPORAL CONSISTENCY

Building upon our success in achieving realistic and semantically consistent transfer of eyeglasses in images, we extend our method to facial videos. Video editing introduces the critical challenge of maintaining temporal consistency, as humans are highly sensitive to discrepancies across frames. Naively applying our image-based method frame by frame results in the eyeglasses appearing slightly different in each frame, leading to a jarring, flickering effect. As illustrated in Fig. 2, we extend the DiffAE-based architecture for video editing by incorporating 3D convolutions and two self-attention layers, specifically spatial-temporal and regional flow-guided self-attention layers, to enhance temporal consistency.

**Extending DiffAE for video editing**   The original U-Net architecture in DiffAE consists of a series of 2D convolutional residual blocks and spatial self-attention blocks. To adapt this architecture for video editing, we inflate the 2D U-Net to a pseudo-3D U-Net to accommodate the temporal dimension, similar to previous works (Cong et al., 2024; Wu et al., 2023). Specifically, we replace each 2D convolutional layer with a pseudo-3D convolutional layer. For a $3 \times 3$ kernel, we adjust it to a $1 \times 3 \times 3$ kernel. We also expand the spatial self-attention blocks to spatial-temporal self-attention blocks, often used in video diffusion models, by using features from the entire video as queries $\mathbf{Q}$, keys $\mathbf{K}$, and values $\mathbf{V}$. However, applying full spatial-temporal self-attention can lead to undesired modifications in regions outside the eyeglasses area as the model attends to irrelevant embeddings. Thus, we introduce a regional self-attention mechanism that uses optical flows to mitigate this problem.

**Regional self-attention with optical flows**   We aim to insert eyeglasses without altering other facial details. To this end, we employ a regional self-attention strategy. While similar local attention has been explored in prior works (Zhang et al., 2022; Cong et al., 2024), we adapt it to our framework by restricting attention to the eyeglass region. In the target video, we define a rough, predefined region of interest (ROI) of the eyeglasses area on the aligned face, computed as the bounding box of the aligned reference mask. We denote the set of all pixels in the ROI bounding box as $\mathcal{C}_{\mathrm{roi}}$, which are appropriately downsampled to fit the spatial dimension of the self-attention block.

We adapt FLATTEN's flow-guided temporal attention to trajectories that pass through our predefined ROI. Instead of attending to the full frame, it attends only to tokens along these ROI-related trajectories, guided by the estimated optical flow. For a video sequence of length $S$, we can obtain a trajectory $\mathcal{T}$ for a pixel in coordinates $(x_0, y_0)$ in the first frame by deriving its coordinates in all subsequent frames as

$$\mathcal{T} = \{(x_0, y_0), ..., (x_s, y_s), ..., (x_S, y_S)\}. \tag{4}$$

We only consider trajectories for which some or all coordinates satisfy $(x_s, y_s) \in \mathcal{C}_{\mathrm{roi}}$, and denote them as $\{\mathcal{T}\}_{\mathrm{roi}} \subset \{\mathcal{T}\}$.

We compute the self-attention of a selected motion trajectory $\mathcal{T} \in \{\mathcal{T}\}_{\mathrm{roi}}$ in a similar manner to FLATTEN (Cong et al., 2024). Let the query $\mathbf{Q}$ of a pixel $(x_s, y_s)$ of frame $s$ be the embeddings $\mathbf{h}(x_s, y_s)$. The corresponding embeddings of the remaining coordinates in the same trajectory, $\mathcal{T}^- = \mathcal{T} - (x_s, y_s)$ are concatenated as

$$\mathbf{H}(\mathcal{T}^-) = [..., \mathbf{h}(x_{s-1}, y_{s-1}), \mathbf{h}(x_{s+1}, y_{s+1}), ...]. \tag{5}$$

Our regional optical-flow-guided self-attention is thus computed using the dimensionality of the embeddings $d$ as

$$\mathbf{Q} = \mathbf{h}(x_s, y_s), \quad \mathbf{K} = \mathbf{V} = \mathbf{H}(\mathcal{T}^-), \quad \mathrm{RA}(\mathbf{Q}, \mathbf{K}, \mathbf{V}) = \mathrm{Softmax}\left(\frac{\mathbf{Q}\mathbf{K}^\top}{\sqrt{d}}\right)\mathbf{V}. \tag{6}$$

By constraining the attention to the eyeglasses region and guiding it with optical flow, we ensure that each embedding in our ROI attends only to its temporally corresponding embeddings across frames. This approach improves the temporal consistency of the eyeglasses while preventing unwanted artifacts in unrelated areas.

## 3.4 OVERALL VIDEO EDITING PIPELINE

We adopt a video editing pipeline proposed in Yao et al. (2021), which comprises three main stages: (1) face alignment and cropping, (2) latent feature encoding, manipulation, and decoding, and (3) unalignment and merging of edited frames into the original video. Given an input video sequence $\mathbf{v} \in \mathbb{R}^{1 \times C \times F \times H \times W}$, where $C$ is the number of channels, $F$ is the number of frames, and $H$, $W$ denote the height and width, respectively, we first align and square-crop the face regions in each frame

and obtain a cropped sequence $\mathbf{x}_{1:F}$ Similarly, we process a reference eyeglasses image to obtain its aligned and cropped face region $\mathbf{x}_{\mathrm{ref}}$. This alignment ensures that the facial features are spatially consistent across frames and with the reference. To enhance temporal consistency in long video sequences that cannot be input at once, we process the video in batches and sample two neighboring frames, one immediately before and one immediately after each batch. These neighboring frames provide additional temporal context, allowing for smoother transitions between batches. The cropped face sequences $\mathbf{x}_{1:F}$ and the cropped reference image $\mathbf{x}_{\mathrm{ref}}$ are then concatenated as $\mathbf{x}_{1:F,\mathrm{ref}}$ and passed through our semantic encoder $\mathrm{Enc}(\mathbf{x}_{1:F,\mathrm{ref}})$. We apply our feature blending strategy and yield the following semantic latent codes:

$$\hat{\mathbf{z}}_{\mathrm{sem}}^{1:F} = \mathrm{Enc}(\mathbf{x}_{1:F}), \quad \mathbf{z}_{\mathrm{sem}}^{\mathrm{ref}} = \mathrm{Enc}(\mathbf{x}_{\mathrm{ref}}). \tag{7}$$

Next, we obtain the stochastic latent representations $\mathbf{z}_{\mathrm{sto}}^{1:F}$ and $\mathbf{z}_{\mathrm{sto}}^{\mathrm{ref}}$ via conditional DDIM inversion using $\hat{\mathbf{z}}_{\mathrm{sem}}^{1:F}$ and $\mathbf{z}_{\mathrm{sem}}^{\mathrm{ref}}$. We construct the blended stochastic latent codes $\hat{\mathbf{z}}_{\mathrm{sto}}^{1:F}$ using $\mathbf{z}_{\mathrm{sto}}^{1:F}$ and $\mathbf{z}_{\mathrm{sto}}^{\mathrm{ref}}$ by Equation (3). The edited frames are then generated by passing the blended stochastic latent codes $\hat{\mathbf{z}}_{\mathrm{sto}}^{1:F}$ and semantic latent codes $\hat{\mathbf{z}}_{\mathrm{sem}}^{1:F}$ through our conditional DDIM model. To ensure temporal coherence, we incorporate regional optical-flow-guided self-attention in all self-attention blocks, applying it exclusively to the target video sequence. Finally, we unalign the edited face regions and paste them back into the original video frames.

## 4 EXPERIMENTS

We evaluate our method with various baselines to confirm its effectiveness. We describe our implementation details and report further experiments in the appendix.

### 4.1 SETTINGS

**Datasets** Since there is no existing dataset tailored for evaluating semantic eyeglass transfer, we construct a benchmark from CelebV-HQ (Zhu et al., 2022). For target facial videos, we randomly select videos without visible glasses that are no shorter than 120 frames, *i.e.*, 4 seconds in 30 fps, and use the first 120 consecutive frames of these videos as target facial videos. To obtain reference eyeglasses, we select videos with clearly visible glasses and apply the following quality filters: an average VSFA score (Li et al., 2019) above 0.8, mean luminance

Table 1: Baseline requirements. Most existing approaches depend on training and/or explicit masks. In contrast, **our method is training-free and target-mask-free**, requiring only a reference eyeglass and the target video.

| Method | Target Mask | Training | Inputs | Backbone |
|---|---|---|---|---|
| TF-ICON (Lu et al., 2023) | Yes | No | Image + Ref + Mask | Diffusion |
| Paint-by-Example (Yang et al., 2023a) | Yes | Yes | Image + Ref + Mask | Diffusion |
| ObjectStitch (Song et al., 2023) | Yes | Yes | Image + Ref + Mask | Diffusion |
| AnyDoor (Chen et al., 2024b) | Yes | Yes | Image + Ref + Mask | Diffusion |
| MimicBrush (Song et al., 2024) | Yes | Yes | Image + Ref + Mask | Diffusion |
| OmniTry (Feng et al., 2025) | No | Yes | Image + Ref | Diffusion |
| DVAE (Kim et al., 2023) | No | Yes | Video + Classifier/Text | DiffAE |
| VideoEditGAN (Tzaban et al., 2022) | No | Yes | Video + Classifier | GAN |
| FLATTEN (Cong et al., 2024) | No | Yes | Video + Text | Diffusion |
| RAVE (Kara et al., 2024) | No | Yes | Video + Text | Diffusion |
| VidToMe (Li et al., 2024) | No | Yes | Video + Text | Diffusion |
| FRESCO (Yang et al., 2024) | No | Yes | Video + Text | Diffusion |
| RF-Solver-Edit (Wang et al., 2025) | No | Yes | Video + Text | Diffusion |
| VACE (Jiang et al., 2025) | Yes | Yes | Video + Ref + Text + Mask | Diffusion |
| **FreeEyeglass (Ours)** | **No** | **No** | Video + Ref | DiffAE |

Ref: Reference Image

$> 90$, head pitch and yaw within $\pm 15°$, and unique identity per sample. After filtering, we randomly selected 100 unique pairs of glasses, comprising 75 eyeglasses and 25 sunglasses, to serve as reference eyeglasses. Finally, we prepare eyeglass and sunglass masks using Grounded-SAM (Ren et al., 2024). Specifically for eyeglass masks, we refine them by subtracting the segmented eye region from the segmented eyeglasses region. We use the aligned mask's bounding box as the ROI for our regional flow-guided self-attention. In total, we construct 100 pairs of target facial videos and reference glasses for our main experiment.

We also render a small-scale CG face dataset of four target identities, each with one pair of eyeglasses and one pair of sunglasses. We render eight ground-truth videos in total for the target identities.

**Baselines** We compare against a wide range of state-of-the-art image and video editing methods. Table 1 summarizes the requirements of all baselines in terms of training, masks, and input modalities. For reference-based image editing, we include TF-ICON, Paint-by-Example, ObjectStitch, AnyDoor, and MimicBrush, which insert a reference object into a background image through inpainting. We also evaluate OmniTry, a concurrent method targeting wearable object try-on. As these are image editing methods, we apply them frame-by-frame to videos.

For video editing, we consider both facial and general text-guided approaches. Facial video editing baselines include Diffusion Video Autoencoder (DVAE) and VideoEditGAN. Text-guided video editing baselines include FLATTEN, RAVE, VidToMe, FRESCO, and RF-Solver-Edit with Hunyuan-Video (Kong et al., 2024) as backbone. We further include VACE 1.3B, a concurrent multi-modal

Table 2: **Quatitative results** on our evaluation benchmark. The top rows show the methods for image editing, while the bottom rows are for video editing. Eye preservation is evaluated on eyeglasses only. Values closer to 1.0 indicate better in TL-ID and TG-ID. **Bold** and underline indicate the best and second-best results. While our method achieves state-of-the-art for many cases, it yields remarkably better *Unified* score $S_{edit}$, balancing the identity preservation and editing quality.

| Methods | Editing Fidelity | | | | Temporal Consistency | | | | Eye Preservation | | $S_{edit}\uparrow$ |
| | $FID_{CLIP}\downarrow$ | $FVD\downarrow$ | CLIP-I↑ | DINO-I↑ | CLIP-F↑ | $E_{warp}\downarrow$ | TL-ID − | TG-ID − | $LPIPS_{eye}\downarrow$ | $SSIM_{eye}\uparrow$ | |
|---|---|---|---|---|---|---|---|---|---|---|---|
| Object Stitch (Song et al., 2023) | 11.626 | 294.12 | 0.882 | 0.624 | 0.957 | 0.0182 | 0.957 | 0.887 | 0.183 | 0.814 | 48.336 |
| TF-ICON (Lu et al., 2023) | 26.394 | 2382.6 | 0.802 | 0.441 | 0.868 | 0.0462 | 0.222 | 0.216 | 0.214 | 0.804 | 17.457 |
| Paint-by-Example (Yang et al., 2023a) | 12.915 | 295.03 | 0.877 | 0.650 | 0.957 | 0.0182 | 0.956 | 0.884 | 0.191 | 0.810 | 48.090 |
| Anydoor (Chen et al., 2024b) | 18.469 | 1007.7 | 0.850 | 0.517 | 0.949 | 0.0279 | 0.779 | 0.748 | 0.242 | 0.781 | 30.452 |
| MimicBrush (Chen et al., 2024a) | 13.399 | 362.31 | **0.909** | **0.739** | 0.959 | 0.0188 | 0.960 | 0.899 | 0.211 | 0.788 | 48.334 |
| OmniTry (Feng et al., 2025) | 11.776 | 415.29 | 0.857 | 0.588 | 0.952 | 0.0173 | 0.892 | 0.684 | 0.183 | 0.813 | 49.670 |
| VideoEditGAN (Xu et al., 2022) | 10.443 | 382.93 | 0.846 | 0.547 | 0.958 | 0.0167 | 0.964 | 0.930 | 0.149 | 0.840 | 50.770 |
| DVAE Classifier (Kim et al., 2023) | 13.624 | 1190.9 | 0.819 | 0.433 | 0.951 | 0.0169 | 0.788 | 0.882 | **0.124** | **0.869** | 48.396 |
| FLATTEN (Cong et al., 2024) | 52.022 | 1522.7 | 0.780 | 0.300 | 0.952 | 0.0154 | 0.860 | 0.718 | 0.175 | 0.839 | 50.809 |
| RAVE (Kara et al., 2024) | 43.152 | 1035.0 | 0.833 | 0.526 | **0.965** | 0.0240 | 0.916 | 0.850 | 0.215 | 0.797 | 34.085 |
| VidToME (Li et al., 2024) | 36.361 | 960.97 | 0.828 | 0.499 | 0.959 | 0.0201 | 0.931 | 0.802 | 0.200 | 0.805 | 41.111 |
| FRESCO (Yang et al., 2024) | 53.046 | 973.35 | 0.787 | 0.486 | 0.952 | 0.0262 | 0.859 | 0.821 | 0.161 | 0.827 | 30.086 |
| RF-Solver-Edit (Wang et al., 2025) | 21.507 | 485.35 | 0.833 | 0.492 | 0.944 | 0.0205 | 0.946 | 0.894 | 0.164 | 0.838 | 40.627 |
| VACE (Jiang et al., 2025) | 9.947 | 223.57 | 0.858 | 0.634 | 0.961 | 0.0167 | **0.974** | **0.942** | 0.163 | 0.836 | 53.136 |
| **FreeEyeglass (Ours)** | **9.839** | **206.37** | 0.865 | 0.542 | 0.962 | **0.0152** | 0.969 | 0.885 | 0.124 | 0.868 | **56.976** |

video editing framework that requires text prompts, target masks, and reference images. The 14B variant of VACE could not be run due to memory limits on A100 80GB GPUs. For text-based baselines, we use GPT-4o (API version 2024-05-13) (OpenAI, 2024) to generate fine-grained descriptions of reference eyeglasses. Implementation details and prompt preparation are provided in the appendix. We use the implementations and pretrained models provided by the authors for all baselines except FLATTEN to generate edited videos. Due to the GPU memory issue, FLATTEN cannot simultaneously process a video of 120 frames, so we split each video into three chunks.

**Evaluation metrics** Since our benchmark lacks ground truth, we assess results using three categories of metrics: *editing fidelity*, *temporal consistency*, and *eye preservation*.

*Editing fidelity:* We use the Fréchet Inception Distance (FID) computed on CLIP features (Kynkäänniemi et al., 2023) to assess overall perceptual quality and Fréchet Video Distance (FVD) (Skorokhodov et al., 2022) to evaluate video realism. We use all frames in the videos for FVD calculation. We compute CLIP-I and DINO-I scores following Wei et al. (2024), using average cosine similarity between each edited frame and the reference eyeglasses image to assess semantic alignment with the reference. CLIP-I uses CLIP ViT-B/32 (Radford et al., 2021), while DINO-I uses DINOv2 ViT-S/16 (Oquab et al., 2024); in both cases, we align frames and crop the eyeglasses region.

*Temporal consistency:* We compute the average cosine similarity between consecutive frame embeddings using CLIP (CLIP-F) to capture feature-level smoothness, and Warp Error (Lai et al., 2018) that measures the pixel-wise difference between adjacent frames warped using the optical flow from the original video. We also compute TL-ID and TG-ID (Tzaban et al., 2022) that quantify identity preservation across adjacent frames or all frames, respectively.

*Eye preservation:* We crop the eyeglasses region for all frames and report $LPIPS_{eye}$ and $SSIM_{eye}$ for the eyeglasses regions of source frames and edited frames, similar to Feng et al. (2025). We conduct the evaluation exclusively on 75 eyeglass pairs, as sunglasses typically obscure the eye region.

We also report a *unified evaluation* score $S_{edit}$, proposed by FLATTEN, defined as the ratio CLIP-I/$E_{warp}$, providing a single metric that balances semantic fidelity and temporal consistency. For the CG face dataset, we compute PSNR, MS-SSIM, LPIPS (Zhang et al., 2018), and MSE between ground truth frames and edited frames generated by ours and the baselines.

### 4.2 RESULTS

**Main results** We present our quantitative results in Table 2 and our visual results compared with our baselines in Fig. 3. We achieve the best unified score $S_{edit}$ among all baselines, which reflects our clear advantage in balancing editing fidelity and temporal consistency. We also achieve superior $FID_{CLIP}$, FVD, and eye preservation scores $LPIPS_{eye}$ and $SSIM_{eye}$, which shows our strength in preserving original video content and overall realism. Qualitative results support these findings. Our method transfers eyeglasses while preserving facial identity and maintaining temporal coherence. Inpainting-based editing baselines reproduce eyeglass details but fail to maintain facial identity. Even the latest works, such as OmniTry and VACE, fail to maintain the eye regions (*e.g.* Eyeglass #1 in Fig. 3). Video editing methods guided by text or classifiers are temporally stable but cannot capture the reference eyeglasses. We provide results for DVAE with text guidance and for the original DiffAE

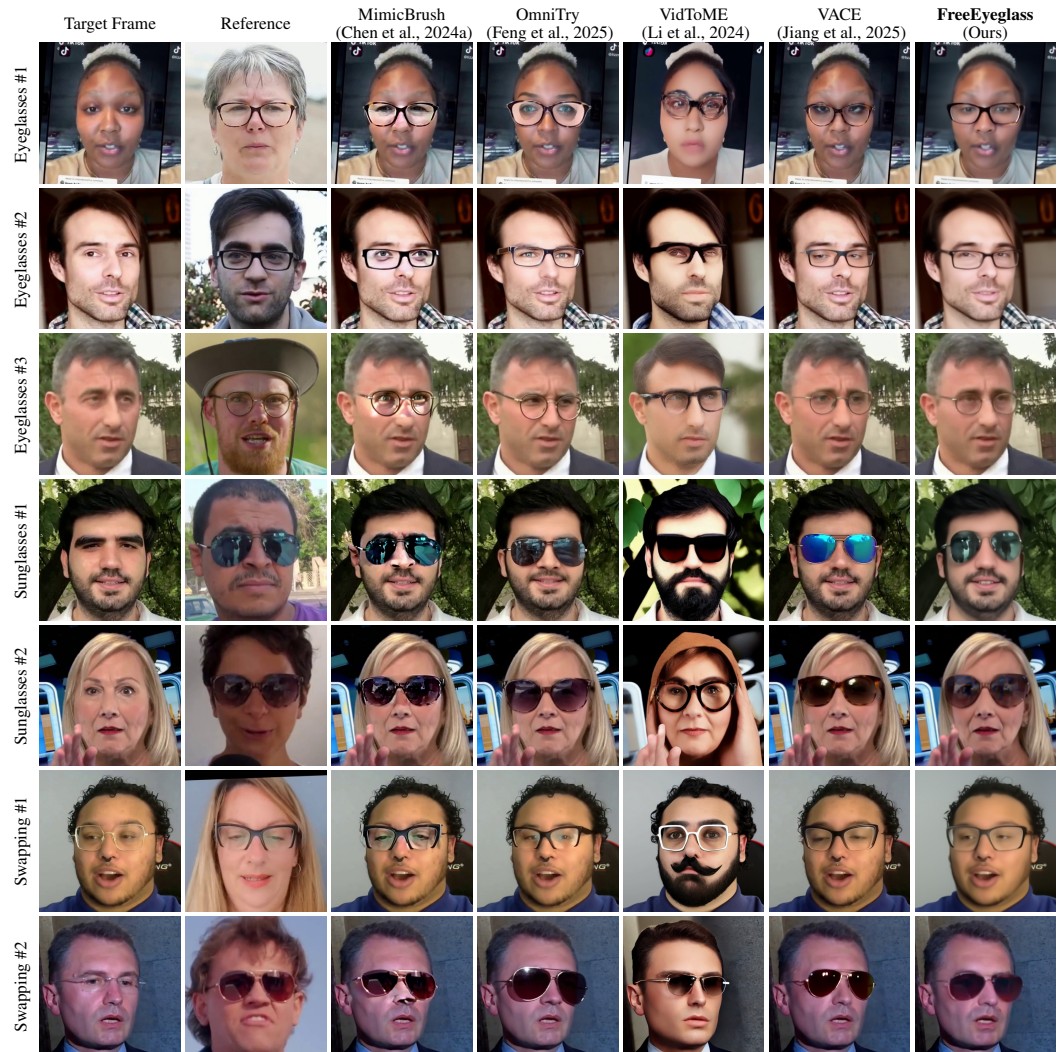

Figure 3: **Visual results** on transferring different eyeglasses compared with our baselines. Our method successfully inserts and swaps eyeglasses from the reference eyeglasses image into the target video, compared to existing baselines. Please refer to the supplementary video for more results.

with classifier guidance for completeness in Sec. C.2. We also include a supplementary video for full comparison.

Our method scores slightly lower on CLIP-I and DINO-I, as these metrics favor methods that closely match the reference, yielding high scores when directly copying eyeglasses. Inpainting-based baselines, including Object Stitch, Paint-by-example, Anydoor, and MimicBrush, use CLIP or DINO features during training, which provides them an advantage. In contrast, our method has no prior exposure to these embeddings. Finally, our framework requires neither per-frame target mask annotation nor model training. Despite being training-free and target-mask-free, it delivers competitive or superior results to training-based state-of-the-art methods.

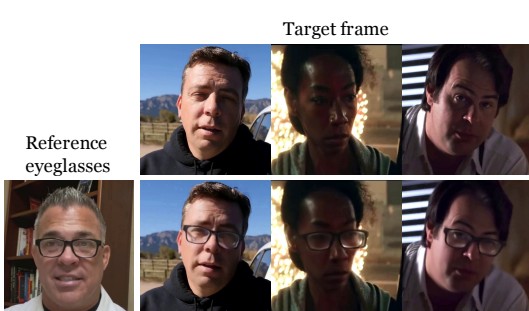

Figure 4: Transferring eyeglasses to target frames under **different illuminations** (*e.g.*, outdoor sunlight, backlit, and side-lit).

**Robustness to lighting variations** Our method remains stable under strong illumination mismatches between the reference eyeglasses and the target video frames. As shown in Fig. 4, it preserves correct eyeglass geometry and placement when transferring from a front-lit indoor refer-

Reference eyeglasses     Target sequence     Edited sequence

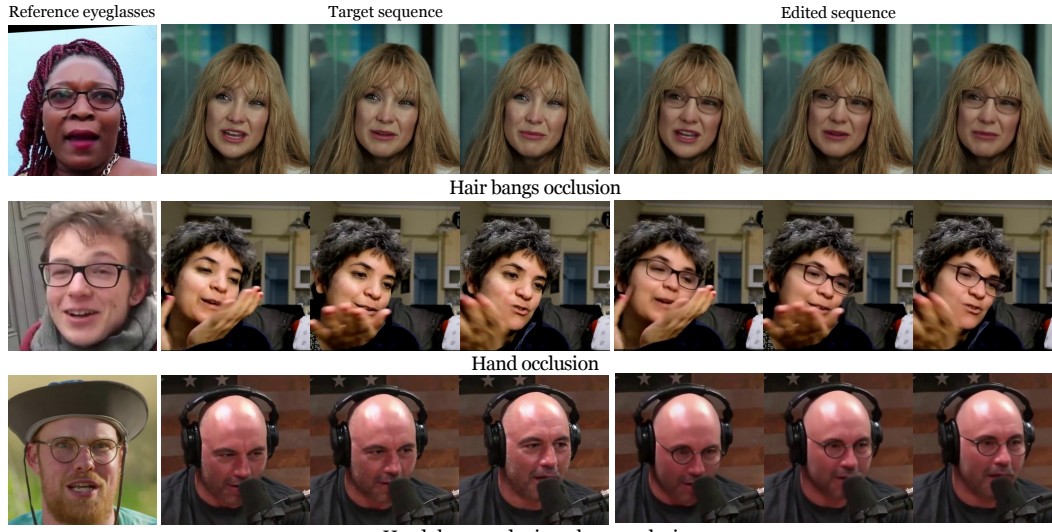

Hair bangs occlusion

Hand occlusion

Headphone and microphone occlusion

Figure 5: **Occlusion handling results.** FreeEyeglass handles cases when the target face is partially occluded. We present three representative occlusions, including hair bangs, hand motion, and object occlusion over the face, where the transferred eyeglasses remain well-aligned and visually coherent.

Target sequence

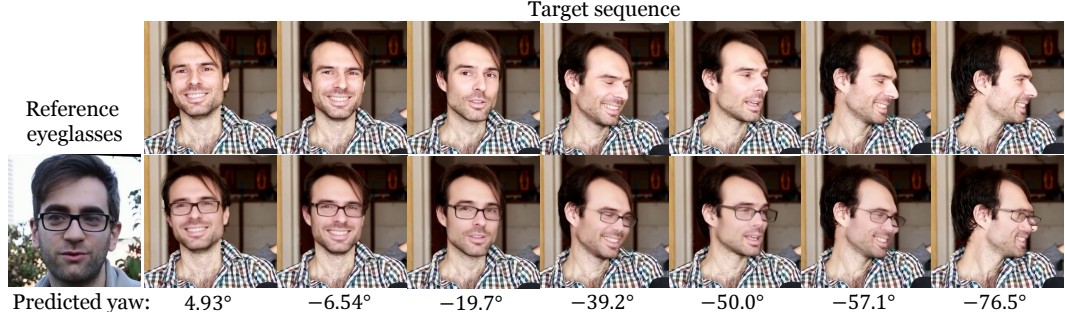

Reference eyeglasses

Predicted yaw:   4.93°   −6.54°   −19.7°   −39.2°   −50.0°   −57.1°   −76.5°

Figure 6: **Eyeglass transfer under head pose variation.** Edited results across increasing yaw angles. Our method remains stable up to roughly ±40–50°, beyond which misalignment and artifacts occur.

ence to targets captured under bright outdoor sunlight, strong back-lighting, and dim indoor side-lit. Across all cases, the transferred glasses integrate cleanly into each scene without introducing artifacts.

**Occlusion handling**     We evaluate FreeEyeglass under several types of partial occlusions. As shown in Fig. 5, the method produces reasonable results in cases involving hair-bang occlusion, dynamic hand motion, and rigid objects such as headphones or microphones. Our method preserves the target identity and eyeglass appearance, while maintaining plausible geometry and placement. We attribute this robustness to the strong reconstruction priors of DiffAE, which enable the model to infer a consistent facial structure despite partial visibility.

**Robustness to pose-change**     We analyze the performance of FreeEyeglass under varying head yaw angles. As shown in Fig. 6, our method remains stable up to approximately ±40–50° of pose change, beyond which the transfer becomes less reliable due to strong geometric mismatch between the reference and target.

**Generalization to other facial attributes**     Although FreeEyeglass is designed for eyeglass transfer, we push the limits of the framework by applying the same mask-only blending strategy to other local facial attributes, including eyebrows, noses, and moustaches (Fig. 7). Without any model modification, the method can inject these attributes into the target sequence with reasonable geometric alignment, which supports that the underlying mechanism extends beyond object category and remains effective for other face-centric edits. We also explore sequential multi-attribute editing and present the results in Sec. C.7.

**Evaluation on CG-rendered ground truth**     We present our quantitative results on CG-rendered scenes in Table 3. Our method outperforms all baselines on PSNR, SSIM, and MSE, and achieves competitive performance on LPIPS, demonstrating superior fidelity across both pixel-level and

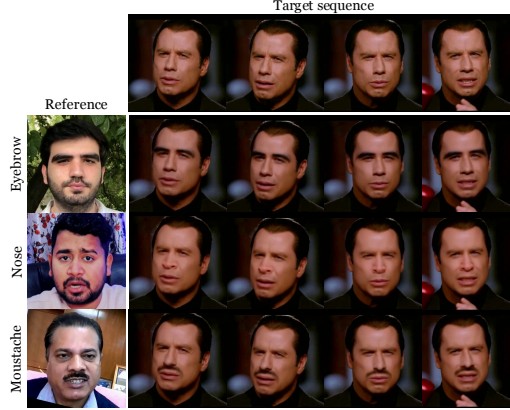

Figure 7: **Transfer of other facial attributes.** FreeEyeglass can transfer additional face-centric attributes (eyebrow, nose, moustache) using the same mask-based blending strategy, without any architectural changes.

Table 3: **Quantitative results** on the CG face dataset. Our method shows leading results on most metrics. **Bold** and underline indicate the best and second-best results.

| Methods | PSNR ↑ | SSIM ↑ | LPIPS ↓ | MSE ↓ |
|---|---|---|---|---|
| Object Stitch | 26.495 | 0.956 | 0.0401 | 481.30 |
| TF-ICON | 10.596 | 0.200 | 0.503 | 17617 |
| Paint-by-Example | 26.482 | 0.957 | 0.0392 | 460.58 |
| Anydoor | 24.457 | 0.948 | 0.0422 | 774.08 |
| MimicBrush | 26.527 | 0.964 | **0.0301** | 467.81 |
| OmniTry | 24.794 | 0.943 | 0.0404 | 679.84 |
| DVAE Classifier | 25.626 | 0.942 | 0.118 | 572.16 |
| VideoEditGAN | 24.008 | 0.912 | 0.0715 | 843.26 |
| FLATTEN | 16.857 | 0.780 | 0.212 | 4245.6 |
| RAVE | 15.942 | 0.799 | 0.175 | 6440.2 |
| VidToME | 17.802 | 0.813 | 0.193 | 3411.6 |
| FRESCO | 18.537 | 0.798 | 0.148 | 2779.3 |
| RF-Solver-Edit | 18.548 | 0.595 | 0.431 | 2936.0 |
| VACE | 26.688 | 0.958 | 0.0375 | 463.53 |
| **FreeEyeglass (Ours)** | **28.425** | **0.967** | 0.0387 | **307.21** |

SSIM: MS-SSIM

Table 4: **Ablation study** on different components of our method. **Bold** and underline indicate the best and second-best results.

| Settings | FVD↓ | CLIP-I↑ | $E_{warp}$ ↓ | TL-ID− |
|---|---|---|---|---|
| Our full model | 206.37 | 0.865 | 0.0152 | 0.969 |
| w/o feat. blend. | **167.65** | 0.819 | **0.0149** | **0.970** |
| w/o blended $\hat{z}_{sto}$ | 280.73 | **0.868** | 0.0162 | 0.946 |
| Per-frame SA | 226.37 | 0.863 | 0.0151 | 0.968 |
| STSA | 226.51 | 0.864 | 0.0151 | 0.968 |
| FlowSA | 223.61 | 0.864 | 0.0151 | 0.969 |
| Regional Per-frame SA | 215.03 | 0.849 | 0.0152 | 0.966 |
| Regional STSA | 193.68 | 0.851 | 0.0152 | 0.958 |

Feat. blend.: feature blending strategy; SA: self-attention
STSA: spatial-temporal self-attention; FlowSA: flow-guided self-attention

| Target Frame | Our full model | Per-frame SA | w/o $\hat{z}_{sto}$ | w/o feat. blend. |
|---|---|---|---|---|

Figure 8: **Visual results** of ablation study. The yellow box denotes the region of temporal inconsistency. Zoom in for a clear comparison.

perceptual metrics. Text-based video editing often over-modifies the appearance and style of the video when applied to local regions, *i.e.*, eyeglass transfer, thus leading to unsatisfactory results. This emphasizes the importance of reference-based editing for achieving precise semantic control.

**Ablation study** We conduct an ablation study across various settings to demonstrate the effectiveness of our method. We ablate our method by switching off different proposed strategies: (1) w/o feature blending, (2) w/o blended stochastic latent $\hat{z}_{sto}$, and different self-attention variants, including per-frame, spatial–temporal, flow-guided, and their regional counterparts. Figure 8 shows that feature blending is essential for inserting the reference eyeglasses; the "w/o feat. blend" variant yields a lower FVD simply because it produces almost no edits and therefore remains closer to the target distribution. Using feature blending without stochastic blending transfers the eyeglasses most faithfully (Table 4), but introduces noticeable noise in Fig. 8. Adding a fixed ROI boundary to per-frame SA or STSA reduces FVD by limiting global interference, but it also lowers CLIP-I because the static ROI often misaligns with the eyeglasses, causing partial corruption of their shapes. In contrast, combining the same ROI boundary with flow-guided attention maintains both semantic correctness and temporal alignment, as it tracks motion trajectories passing through the ROI and keeps the attended features aligned with the moving eyeglasses across frames. We also compare different locations to apply feature blending within DiffAE (see Sec. C.1). Results confirm that applying blending in the semantic encoder yields the best identity-preserving edits, while other locations cause artifacts or copy-paste effects.

## 5 CONCLUSIONS

This paper has focused on naturally adding and replacing eyeglasses on faces in videos. To this end, we introduce a feature blending strategy and a regional flow-guided self-attention mechanism. Experiments show that our method successfully transfers the reference eyeglasses into the target video, creating harmonized results that accurately reflect the original eyeglasses. Meanwhile, our method faces challenges when swapping eyeglasses that differ significantly in style or completely removing eyeglasses from an image (see Sec. G). This is because fine details of the eyeglasses that the semantic encoder fails to capture are embedded into the stochastic latent space, making them difficult to manipulate. Improving the semantic encoder to better capture eyeglass semantics may address this issue.

ETHICS STATEMENT

This work uses the CelebV-HQ dataset, which is publicly released and collected from YouTube, following the dataset's license and usage policy. We also employ a CG dataset created from human scans with informed consent; these data are used solely for research purposes and are kept private. Additionally, we conduct a user study, but we do not collect any personally identifying information from participants. No human subjects were recruited or recorded directly by the authors. We acknowledge potential risks of misuse, such as in deepfake generation, and limit our contributions to academic research with no intent for malicious applications.

REPRODUCIBILITY STATEMENT

We provide implementation details, hyperparameter settings in Sec. B, and dataset preprocessing steps in Sec. 4.1. We will release our source code and relevant scripts upon acceptance.

LLM USAGE DISCLOSURE

We used a large language model (LLM) to polish the writing style and grammar in parts of the manuscript. The LLM did not produce novel scientific content or technical contributions. All statements and claims are verified and edited by the authors. We take full responsibility for all content.

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
