# OpenReview forum: "FreeEyeglass: Training-free and Mask-free Eyeglass Transfer for Facial Videos"
_ICLR.cc/2026/Conference — Submitted to ICLR 2026_

### Official Review · Reviewer_xwtJ · 2025-10-30

**Soundness:** 3
**Presentation:** 3
**Contribution:** 3
**Rating:** 8
**Confidence:** 5

**Summary:**

This paper presents a method to transfor eyeglasses in videos of faces using the Diffusion Autoencoder framework. The method first employs a feature blending step, using an input mask of the glasses, followed by a "regional self-attention", based on optical flow, in to order to target the region to put the glasses in.

**Strengths:**

I found the paper well-written, quite easy to follow, the different steps makes a lot of sense, and the final results are pretty good, with good temporal consistency. I am very happy with the extensive comparisons with the state-of-the-art.

**Weaknesses:**

One of the main criticsims I would have of the paper is that the application seems extremely "small" compared to the sophisticated nature of the method. Did you try to put other objects rather than eyeglasses ? Does this work ? I think one really would expect other examples for such a seemingly powerful method. If you wish to keep the glasses application only, then you should justify this (at least to the reviewers). I feel that in an actual applicative (industrial) context, that the tool would be extremely limited, a user is not going to want to load a different model for each type of object to insert.

Another point is that the region self-attention is proposed as novel, and a contribution of yours. But then you basically say after that you use the method of "FLATTEN: optical FLow‑guided ATTENtion for consistent text‑to‑video editing", Cong et al 2022. Please make it clearer whether you are just using their method, or if there is some difference between the two ? Otherwise, I think the idea makes a lot of sense.

**Questions:**

One of the main criticsims I would have of the paper is that the application seems extremely "small" compared to the sophisticated nature of the method. Did you try to put other objects rather than eyeglasses ? Does this work ? I think one really would expect other examples for such a seemingly powerful method. If you wish to keep the glasses application only, then you should justify this (at least to the reviewers). I feel that in an actual applicative (industrial) context, that the tool would be extremely limited, a user is not going to want to load a different model for each type of object to insert.

Another point is that the region self-attention is proposed as novel, and a contribution of yours. But then you basically say after that you use the method of "FLATTEN: optical FLow‑guided ATTENtion for consistent text‑to‑video editing", Cong et al 2022. Please make it clearer whether you are just using their method, or if there is some difference between the two ? Otherwise, I think the idea makes a lot of sense.

How is the mask M obtained ? Do you have an eyeglasses segmentation network or something similar ? If so, how to you resize the mask to fit each layer of the U-Net. This seems to be quite an important point, as the blending process hinges on this.

Equation 2: you add a binary mask to a blurred version of itself, and then clamp the output values. This looks a bit weird at first, but by doing this it seems that you keep the inner parts of the "1" regions of the mask to 1, and just blur the borders a bit, is this true ? If so, maybe you could explain it like this. In other words, you want a smooth blending on the border, but in the inner part, the blending should be binary. It is not super well explained why this is the best choice. This is done somewhat at the end of the paragraph, but I did not find the explanation convincing, for example why should this blending choice imrove "the preservation of fine details in the edited images", it just seems to make a smoother border transition no ?


Specific details:

- p. 5, line 239: "We implement the proposed regional self-attention concept to the state-of-the-art temporal attention using optical flows". Rephrase, sentence is not comprehensible
- section 3.3: local self-attention is a pretty well-studied subject, you present it as if there are no other works on it. In particular:
- "Flow‑Guided Transformer for Video Inpainting", Zhang et al, ECCV 2022
- "FLATTEN: optical FLow‑guided ATTENtion for consistent text‑to‑video editing" (as you cite)
- small note: when one says "eyeglasses", it usually means as opposed to sunglasses; but your method does both. Maybe change this (although this is not crucial)

---

> ### Author Response · Authors · 2025-11-17
>
> Thank you very much for your thoughtful and encouraging review.
> We sincerely appreciate your detailed reading and constructive questions.
>
> ### W1. & Q1. Scope of Application
> Our current study focuses on eyeglasses because they pose unique challenges: they occlude identity-critical regions and require precise geometric alignment.
> From our experiments, we observed that the pre-trained DiffAE semantic encoder best preserves fine-grained details within the facial area, likely reflecting its training prior (FFHQ dataset). Within this domain, the framework behaves **object-agnostically**; it can handle different facial features or accessories, such as sunglasses, masks, eyebrows, or beards, simply by providing the corresponding reference mask.
> We will include several of these examples in the revision.
> Extending the approach to accessories outside the face region (e.g., hats) would mainly require adapting or retraining the base encoder to a wider spatial domain, which we consider a promising direction for future work.
>
>  &nbsp;
> ### W2. & Q2. Our Proposed Regional Self-attention
>
> We adopt the optical-flow guidance idea from FLATTEN but apply it only **within a bounded region**, which is derived from the eyeglass mask’s bounding box. This limits cross-frame interference and avoids artifacts we observed when global flow-guided attention is used. We will clarify this distinction and revise Sec. 3.3.
> We thank the reviewer for pointing out the unclear sentence on p.5, line 239, and we will rephrase it for clarity. Specifically, we will revise the wording to *“We adapt FLATTEN’s flow-guided temporal attention to trajectories that pass through our predefined ROI. Instead of attending to the full frame, it attends only to tokens along these ROI-related trajectories, guided by the estimated optical flow.”*
>
> We also appreciate the reviewer’s pointer to Flow-Guided Transformer for Video Inpainting (ECCV 2022) and will cite it as relevant prior work on local attention.
>
> &nbsp;
> ### Q3. Mask Preparation
>
> We obtain the eyeglass mask automatically using Grounded-SAM. We refine it by subtracting the segmented eye region and applying light morphological operations when necessary. The mask is bilinearly downsampled at each ResBlock of the semantic U-Net to match feature resolution. We will describe these details explicitly in the revision.
>
> &nbsp;
> ### Q4. Equation 2 and the stochastic latent feature blending
>
> We appreciate the reviewer’s careful reading. Our previous wording for stochastic latent feature blending (“introducing the details of the reference eyeglasses”) was imprecise. Empirically, we observe that directly using the original stochastic latent often **introduces local noise and border artifacts** (see Fig. 5). By smoothing the border of the blending mask, we stabilize the transition between target and reference features while keeping the inner region binary to preserve fine detail. We will revise the text to emphasize that this operation primarily **stabilizes the latent features and reduces artifacts**.
>
> &nbsp;
> ### Q5. Terminology
> Thank you for noting the terminology. We used “eyeglasses” in a broad sense that includes sunglasses; we will clarify this usage in the revision.
>
>
> Thank you again for your positive assessment and for highlighting the key strengths of our work. We hope that this reply answers your questions.

---

> ### Author Response · Authors · 2025-12-02
> **Summary of Revisions in Response to Reviewer xwtJ**
>
> Dear Reviewer,
>
> Thank you very much for your thoughtful review and for the strong positive assessment of our method, clarity, and experimental evaluation. We have carefully updated the manuscript to address all of your comments. Below, we summarize the corresponding revisions.
>
> * **Broader applicability beyond eyeglasses** (W1& Q1)
>
>     To address the concern about the application scope, we added examples for eyebrows, nose, and moustache transfer in Sec 4.2 and a sequential editing example (nose → eyeglasses) in Appendix C.7. These results confirm that the framework is not task-specific and supports broader facial attribute editing without architectural changes.
>
> * **Clarification of regional self-attention & related work** (W2 & Q2)
>
>     We refined the explanation of regional flow-guided self-attention and explicitly distinguished our adaptation from prior work. We also added citations to related approaches as suggested (Sec. 3.3).
>
> * **Mask preparation and usage** (Q3)
>
>     We clarified how the reference mask is obtained and used in the pipeline:
>      * Sec. 3.2 now states that the reference mask is bilinearly downsampled to latent resolution.
>      * Sec. 3.3 explains how the aligned reference mask defines the ROI for target frames.
>
> * **Stochastic latent feature blending** (Q4)
>
>     We revised Sec. 3.2 to clarify that stochastic latent blending stabilizes target–reference transitions and reduces artifacts, and that smoothing only the mask boundary preserves this effect while retaining details in the interior.
>
> We sincerely appreciate your constructive feedback and are grateful for your positive evaluation of our work. We hope the revised manuscript fully addresses your concerns.
>
> Best regards,
>
> Authors

---

### Official Review · Reviewer_NpfN · 2025-10-31

**Soundness:** 2
**Presentation:** 2
**Contribution:** 2
**Rating:** 2
**Confidence:** 4

**Summary:**

This paper aims to solve the problem of eyeglass transfer in facial videos. This paper proposes a "training-free" and "mask-free" framework built upon a pre-trained Diffusion Autoencoder (DiffAE). The core method involves blending reference features in the DiffAE's semantic encoder while also fusing stochastic latents to preserve details . To handle videos, the method extends the 2D architecture to pseudo-3D and employs a regional, flow-guided self-attention mechanism to maintain temporal consistency .

**Strengths:**

1. The paper addresses a commercially relevant and technically challenging problem (video virtual try-on) that is currently underexplored, especially compared to virtual clothes try-on.
2. The choice of a Diffusion Autoencoder (DiffAE) as the backbone is well-motivated. For a local editing task like this, the explicit reconstruction objective of an autoencoder is crucial for preserving unedited facial regions and identity, a key insight of the paper .

**Weaknesses:**

1. The "mask-free" is over claimed. While the target video does not require a mask, the method explicitly requires a binary mask $M$ for the reference image. This mask, obtained from Grounded-SAM, is fundamental to the feature-blending operation.
2. limited novelty. The method is largely an assemblage of existing techniques. The backbone is DiffAE , the video editing pipeline is borrowed from prior work , and the temporal attention mechanism is a direct adaptation of FLATTEN.
3. Narrow evaluation.The evaluation benchmark is strictly filtered, limiting videos to small head poses ($\pm15^{\circ}$). This avoids the most difficult challenges of eyeglass transfer, such as maintaining 3D consistency during large-angle rotations, which is critical for a robust real-world application.

**Questions:**

1. The compared baselines are general try-on or only for eyeglasses？It may be unfair to compare to general try-on methods.
2. Is the method robust to the eye masks? An analysis of how inaccuracies in the mask itself (i.e., the issue of robustness) would affect the model's performance is necessary.

---

> ### Author Response · Authors · 2025-11-17
> **Clarification on Evaluation Setup and Baselines**
>
> Thank you very much for your detailed review and constructive feedback. We would like to clarify two points that may have caused confusion:
>
> ### 1. Benchmark scope (W3)
> We apologize for the confusing wording. Our evaluation benchmark **does not restrict target videos** to small head poses (±15°) as mentioned in the review. The dataset includes target videos with **unconstrained** head poses, expressions, and lighting variations. The only filtering criterion applies to the reference images, which must display clearly visible eyeglasses so that transfer quality can be evaluated consistently.
> We will clarify this point in Sec. 4.1 of the revision.
>
> ### 2. Baselines (Q1)
> We compare with OmniTry, which is specifically trained for eyeglass try-on. As there are no other publicly available eyeglass-specific try-on baselines, we include object-insertion methods that are theoretically applicable to eyeglass transfer. We do not include general clothing try-on systems, which are usually trained for full-body garments and are not designed for localized facial editing. Note that our framework is training-free, operating directly on a pre-trained Diffusion Autoencoder, while most baselines require task-specific training.
>
> We appreciate the reviewer’s further questions on novelty and robustness, which we will address in a later comment with additional analysis and clarification.

---

> ### Author Response · Authors · 2025-11-20
>
> We appreciate the reviewer’s thoughtful comments and the positive remarks on the practical importance of video try-on and the motivation behind using DiffAE for reference-guided editing. We address the remaining concerns on novelty, robustness, and methodological claims below.
>
> **W1. On the “mask-free” claim**
>
> Thank you for pointing this out. Our method is mask-free on the target video; no per-frame target annotations or masks are required. We acknowledge that the original wording was ambiguous and will revise it to “target-mask-free” for clarity. The single reference mask is obtained automatically via Grounded-SAM and does not involve any manual labeling.
> Obtaining a per-frame target mask is extremely impractical for eyeglass transfer. Since the eyeglasses are *absent* in the target sequence, their position, geometry, and occlusion relationships cannot be reliably annotated or predicted across hundreds of frames. Therefore, target-mask requirements would make real-world use cases infeasible, and supporting a target-mask-free pipeline is essential for practical video try-on scenarios. We will clarify this motivation in the revision.
>
>
> **W2. On the novelty of the method**
>
> While our pipeline incorporates components inspired by prior work, the contribution is not a direct assembly of existing techniques. Our main novelty lies in introducing a new editing paradigm: reference-guided editing within a reconstruction-based diffusion autoencoder, enabled by a key scientific observation about the latent space of DiffAE.
>
> Through extensive experiments, we found that DiffAE’s semantic latent space provides not only appearance preservation but also *geometric and positional alignment*. Remarkably, even when given only a coarse, face-aligned reference mask, without any explicit pose modeling, keypoints, 3D warping, or target masks, the model adapts the reference eyeglasses to the target face. It corrects their placement, orientation, and visible geometry so that the output naturally appears as if the target person is wearing the eyeglasses. To our knowledge, this semantic alignment property has not been documented in prior works.
>
> We believe this finding is *scientifically meaningful* and goes beyond a simple combination of existing components, as it reveals an unexpected capability of reconstruction-based diffusion autoencoders.
> In addition, our training-free and target-mask-free editing framework, built upon this insight, provides a practical and technically significant contribution that further distinguishes our approach from inpainting-based editing methods. We will clarify these points more explicitly in the revised version.
>
>
> **Q2. On robustness to mask inaccuracies**
>
> Thank you for raising this point. In practice, our method is robust to moderate mask inaccuracies. In our experiments, even coarse or *hand-drawn* reference masks consistently produce visually reasonable transfer results. We are preparing a set of examples illustrating robustness to mask variation, and will include these in the revision.

---

> ### Author Response · Authors · 2025-12-02
> **Summary of Revisions in Response to Reviewer NpfN**
>
> Dear Reviewer,
>
> We sincerely thank you for your thoughtful review and constructive feedback. We have carefully revised the manuscript and supplementary materials to address all of your concerns. Below, we summarize the key updates corresponding to your comments:
>
> * **Clarification of “mask-free” terminology** (W1)
>
>     We revised the “mask-free” terminology throughout to target-mask-free to avoid the earlier ambiguity (title and main text).
> * **Novelty and contribution** (W2)
>
>     We strengthened the introduction to highlight our core empirical insight: DiffAE’s semantic latent space naturally supports geometric and positional alignment. This scientific observation underpins our method and goes beyond a direct assembly of prior techniques.
> * **Evaluation benchmark clarification** (W3)
>
>     We revised Sec. 4.1 to clarify that target videos are not restricted. The filtering applies only to the reference images to ensure that eyeglasses are clearly visible for evaluation. The section was restructured to avoid ambiguity.
>
> * **Robustness to mask inaccuracies** (Q2)
>
>     We added a new robustness experiment (Appendix C.5) demonstrating that our method remains stable with dilated, shifted, and rough hand-drawn reference masks.
>
>
> We trust that the revisions and clarifications provided herein adequately address your concerns. We are grateful for your thoughtful feedback and the opportunity to improve the manuscript.
>
> Best regards,
>
> Authors

---

### Official Review · Reviewer_r74s · 2025-10-31

**Soundness:** 3
**Presentation:** 2
**Contribution:** 2
**Rating:** 2
**Confidence:** 4

**Summary:**

This paper introduces FreeEyeglass, a training-free framework for reference-guided eyeglass transfer in facial videos. It leverages Diffusion Autoencoders (DiffAE) for editing purpose, which is claimed to be better at perceiving local identities than the inpainting methods.

The key innovations include (1) feature blending in the semantic encoder and stochastic latent to inject reference eyeglass information and (2) a regional self-attention guided with optical flows to edit only desired regions in input videos.

**Strengths:**

1. The method is training-free, which makes it lightweight and easy to deploy compared to approaches requiring fine-tuning or retraining.
2. The proposed feature and stochastic blending strategy are well-motivated and technically reasonable.  Inflating the 2D method to 3D is interesting, and a regional self-attention with optical flows is proposed to ensure better preservation of undesired regions in the input video.
3. The evaluation covers multiple baselines and various metrics. The reported metrics and quantitative results demonstrate improved performance on the eyeglass transfer task.

**Weaknesses:**

1.	The claim that the method is mask-free is somewhat misleading. It still requires a segmentation mask for the reference eyeglasses and a predefined region of interest (ROI) in the target video.
2.	The scope of the paper feels narrow. While eyeglass transfer is an interesting and useful application, the method itself could likely generalize to other types of local edits beyond eyeglasses, given it is training-free and not inherently task-specific (aside from the eyeglass mask).
3.  From the ablation study (Table 4), removing one of the blending might result in better results in certain metrics. This is different from what we observed in the qualitative results from Figure 5. This raise my question about the effectiveness of the proposed blending strategy, or the soundness of selected metrics (as they should align with qualitative results).
4. The work is build on DiffAE, where the original method demonstrated the possibility of interpolating the semantic latents for editing. This work extend the editing mechanism by blending the video features with masked glass feature -- the change seems to be incremental.
5. The effectiveness of regional self-attention should also be ablated.
6. As shown in the supplement material, the temporal consistency achieved is still somewhat limited, and artifacts can appear across frames.

**Questions:**

1.  How well does the approach handle occlusions (e.g., hair, hands) or extreme lighting variations during eyeglass transfer?
2.	Could the feature blending mechanism be extended to other accessories (e.g., hats, earrings) or to multi-object editing scenarios?
3.	How sensitive is the model to inaccuracies or misalignments in the reference mask?
4.  What happens if the reference viewpoint is significantly different from the target (e.g., reference captured from the left side and target from the right)?
5. How does the editing performance compare to the original DiffAE when classifier guidance (e.g., a glasses classifier) is incorporated?

---

> ### Author Response · Authors · 2025-11-20
>
> We thank the reviewer for the constructive feedback and for recognizing the strengths of our training-free design, technical contributions, and evaluations. We address all concerns and questions below.
>
> **W1. On the “Mask-free” claim**
>
> Thank you for pointing this out. Our method is *target-mask-free*, meaning that no per-frame masks or annotations are required for the target video. We agree that the original wording was ambiguous, and we will revise it to better reflect this intended meaning.
> Obtaining a per-frame target mask is extremely impractical for eyeglass transfer, as the eyeglasses are absent in the target sequence, making it impossible to reliably annotate or infer their precise position, shape, and occlusion relationships across multiple frames. This makes target-mask requirements unsuitable for real-world applications. Our pipeline, therefore, relies only on a single reference mask, which can be obtained automatically using Grounded-SAM followed by the lightweight refinement described in Sec. 4.1.
>
> **W2. Scope of the method**
>
> We appreciate the reviewer’s observation. While our method is training-free and not inherently task-specific, we intentionally focus this work on eyeglass transfer because this application presents unique challenges compared to other facial edits. Eyeglasses occlude identity-critical regions, exhibit complex geometry, and require fine-grained geometric alignment across video frames. These factors make eyeglass transfer significantly more difficult than other facial edits and justify studying it as a dedicated problem.
> At the same time, we agree with the reviewer that the underlying mechanism is not limited to eyeglasses. Because the framework operates on localized latent regions rather than object categories, it naturally extends to other face-centric attributes. We have verified preliminary success on medical masks, eyebrows, and beards using only the corresponding reference mask, without any retraining. We will include representative examples in the revised version.
>
>
> **W3. The ablation results and metric interpretation**
>
> We appreciate the reviewer’s careful reading of Table 4. The behavior of the metrics is actually consistent with the qualitative results. When feature blending in the semantic encoder is removed, the model produces almost no visible edit, as also shown in Fig. 5. Because FVD compares the edited video against the target video distribution, a “no-edit” output naturally yields a numerically lower FVD, even though it is qualitatively undesirable. This explains why the “w/o feat. blend” variant may appear better under FVD despite failing to insert the eyeglasses.
> For the stochastic latent blending, our original description imprecisely stated its role in “introducing details of reference eyeglasses.” Empirically, using the unblended stochastic latent tends to introduce local noise and boundary artifacts, while the proposed boundary-smoothed blending stabilizes the latent transition and reduces such artifacts. These artifacts affect CLIP-I only weakly because CLIP-I captures global semantic similarity, not local structural smoothness. This explains the slightly higher CLIP-I of the “w/o blended $\hat{\mathbf{z}}_{\text{sto}}$” variant despite its lower visual quality in Fig. 5.
> We will revise Sec. 3.2 to clarify the design motivation of the stochastic blending and add a note discussing the interplay between metrics and qualitative behavior.
>
>
> **W4. The relationship to DiffAE and incremental novelty.**
>
> We acknowledge that DiffAE supports global semantic interpolation (e.g., toggling “with/without eyeglasses”). However, this mechanism performs only attribute-level editing and does not support reference-specific transfer or temporally consistent manipulation.
>
> While our method introduces a simple feature-blending strategy to enable reference-based editing, the key novelty of our work is *scientific rather than purely technical*. Through extensive experiments, we found that DiffAE’s semantic latent space exhibits an unexpected capability for *geometric and positional alignment*. Even when given only a coarse, face-aligned mask for the reference eyeglasses, without any pose modeling, keypoints, 3D warping, or target masks, the model adapts the reference eyeglasses to the target face, correcting their placement, orientation, and visible geometry. To our knowledge, this semantic alignment behavior has not been previously documented.
>
> This scientific finding motivates a new editing paradigm: training-free, target-mask-free, reference-guided video editing within reconstruction-based diffusion autoencoders. While the technical mechanism enables this editing, it is the underlying insight into the latent space that fundamentally distinguishes our approach from prior DiffAE editing or inpainting-based pipelines. We will make this scientific contribution clearer in the revision.

---

> ### Author Response · Authors · 2025-11-20
> **(Cont.) Official Comments of Authors**
>
> **W5. The ablation of regional self-attention.**
>
> Thank you for pointing this out. In Table 4, we have already ablated the attention mechanisms, ImgSA (per-frame self-attention), STSA (spatio-temporal self-attention), and FlowSA (flow-guided temporal attention). Our full model uses FlowSA within the bounded region, i.e., regional FlowSA.
> We agree that an additional ablation isolating the regional component itself (e.g., ImgSA/STSA restricted to the bounded region but without flow guidance) would make the study more complete. We are currently running these variants and will include them, along with a discussion, in the revised version.
>
> **W6. Temporal inconsistency and artifacts.**
>
> We appreciate the reviewer’s observation. Our method is built on a pre-trained image diffusion autoencoder and therefore does not benefit from a learned video latent space. In addition, our temporal self-attention relies on estimated optical flow, whose inaccuracies naturally limit the achievable consistency in any training-free setting.
> Despite this, our approach achieves the best overall realism (FVD, $FID_{\mathrm{CLIP}}$ ) and leading temporal consistency (2nd in CLIP-F and TL-ID, 1st in  $\mathrm{E}_{warp}$ ) among all baselines, including video-editing methods trained directly on videos (Table 2). This indicates that the proposed mechanism already provides strong temporal stability compared to existing methods.
> We will improve the explanation in the revised version for clarity.
>
>
> **Q1. Handling occlusions or extreme lighting variations**
>
> In our tests, the method performs reliably when the face is partially visible or when occlusions such as headphones or microphones are present. Our evaluation benchmark also includes cases where the target video is recorded under lighting conditions that differ noticeably from the reference image. In our tests, the method handles such illumination differences reasonably well. We are evaluating additional occlusion and more challenging lighting scenarios and will include representative examples in the revision.
>
> **Q2. Extending the blending mechanism to other accessories or multi-object editing**
>
> In principle, the proposed blending mechanism is not restricted to eyeglasses, and our preliminary experiments suggest it also supports several face-centric attributes. We are currently evaluating multi-attribute editing scenarios and will include the corresponding examples in the revision.
>
> **Q3. Sensitivity to mask inaccuracies**
>
> We have observed that our method produces plausible results even with *hand-drawn* or coarse reference masks. We are preparing a set of examples illustrating robustness to mask variation, and will include these in the revision.
>
> **Q4. Viewpoint differences**
>
> In our experiments, the method handles viewpoint differences. For example, the teaser (Figure 1) shows a left-facing reference face (upper row), while the target video (left) undergoes noticeable pose changes and is predominantly facing the upper right; the transfer remains consistent throughout the sequence. This suggests that the semantic encoder can adapt reference eyeglasses to a range of target poses.
> We are currently preparing additional examples with a larger viewpoint difference and will include them in the revision.
>
> **Q5. DiffAE with classifier guidance**
>
> Thank you for the suggestion. Classifier guidance in DiffAE operates at the attribute level (e.g., “wearing glasses” vs. “not wearing glasses”) rather than preserving the appearance of the reference eyeglasses, and it also does not provide temporal consistency for videos. Because of this, it is not directly suited for reference-guided transfer. Nevertheless, we are running experiments with DiffAE + classifier guidance to quantify the difference and will include the results in the revision.

---

> ### Author Response · Authors · 2025-12-02
> **Summary of Revisions in Response to Reviewer r74s**
>
> Dear Reviewer,
>
> We sincerely thank you for the detailed assessment and many constructive suggestions. We have carefully revised the manuscript and supplementary materials to address all the raised concerns. Below is a summary of the updates corresponding to your comments:
>
> * **Clarification of the “mask-free” terminology** (W1)
>
>     We revised the wording throughout the paper to target-mask-free (title + main text). We clarify that the method only requires a single automatically obtained reference mask, and we now explicitly describe how the target ROI is derived from the aligned reference mask (Sec. 3.3).
>
> * **Scope of the method & broader applicability** (W2, Q2)
>
>     We clarified why eyeglass transfer is a particularly challenging and representative case. At the same time, we demonstrate that the mechanism is not eyeglass-specific by adding new results on eyebrows, noses, and moustaches (Sec. 4.2), as well as a sequential nose→eyeglasses editing example (Appendix C.7).
>
> * **Ablation results and metric interpretation** (W3)
>
>     We added an explanation reconciling FVD/CLIP-I behavior with qualitative results (Sec. 4.2), and we refined Sec. 3.2 to clarify the role of stochastic latent blending in reducing boundary artifacts.
>
> * **Novelty relative to DiffAE & classifier guidance** (W4, Q5)
>
>     We revised the introduction to clarify our key empirical insight on DiffAE’s geometric alignment capability. We added a comparison with DiffAE + classifier guidance (Appendix C.2), confirming its limitations in terms of reference fidelity and temporal consistency and explaining how our reference-guided editing differs from DiffAE’s original mechanism.
>
> * **Ablation of regional self-attention** (W5)
>
>     As suggested, we added ablations of regional Per-frame SA and regional STSA (Sec. 4.2). These confirm that our flow-guided regional SA yields the best preservation of eyeglass context and temporal consistency.
>
> * **Robustness to challenging scenarios** (Q1, Q3, Q4).
>
>     We added new robustness figures addressing all three questions:
>     * **Lighting & occlusions** (Q1):
>
>         New examples under direct sunlight, backlit and side-lit conditions, as well as hair/hand/object occlusions (Sec. 4.2).
>     * **Mask inaccuracies** (Q3):
>
>         Appendix C.5 shows robustness under dilated, shifted, and rough hand-drawn masks.
>     * **Viewpoint mismatch** (Q4):
>
>         Appendix C.6 shows plausible transfer with large viewpoint differences (left→right, pitch mismatch).
>
> * **Temporal inconsistency and artifacts.** (W6)
>
>     We included a discussion on suppressing further minor artifacts in the Appendix.
>
>
> We hope that these updates and our clarifications clearly address all your concerns. Thank you again for your thoughtful review and valuable guidance.
>
> Best regards,
>
> Authors

---

### Official Review · Reviewer_f7Lq · 2025-11-03

**Soundness:** 3
**Presentation:** 2
**Contribution:** 2
**Rating:** 4
**Confidence:** 4

**Summary:**

In this paper, the authors present a novel method for virtual eyeglass try-on that employs off-the-shelf generative models and requires neither training nor fine-tuning. The method outperforms many image and video editing algorithms that are not tailored to eyewear applications. The authors argue that this comparison is fair, as their methodology does not require additional training and therefore should be compared to general visual editing models. The proposed method is fairly robust to reference-target mismatches and slight head pose changes, and it performs well in terms of temporal consistency. Overall, the results are impressive, and the paper paves the way for future research in virtual glasses try-on by providing a benchmark dataset and suitable metrics for eyeglass transfer to video.

**Strengths:**

### Originality

The proposed method is straightforward, building on the cut-overlay-blend pipeline, but is nevertheless original in its use of noise-latent-space editing and Diffusion Autoencoders with temporal self-attention, which together provide improved conformity to target head poses and better video consistency.

### Quality

The paper adheres to scientific standards, providing good ablation studies and comprehensive comparisons.

The methodology is interesting and provides a strong baseline for this task.

### Clarity

The paper is easy to follow, and most ideas are described thoroughly, perhaps with an excessive mathematical formalism, but this does not hurt readability.

### Significance

The focus on eyewear transfer is likely too narrow, as the proposed method can be generalized to support a broader range of wearables and facial features, such as moustaches, jewellery, or headsets, etc. Therefore, the significance could be rated higher, since there is great potential for wider practical applications.

**Weaknesses:**

### What is Missing

 * The authors do not present results for more extreme head poses (e.g., significant rotations relative to the reference image). The supplementary material shows that differences in scale, such as head width, are not well handled. The videos provided are short and nearly static, so the model's behavior in cases of abrupt angle changes, occlusions (e.g., a hand touching the face or hair obscuring the eyeglasses), or rapid movement is not demonstrated. While one could argue that try-on conditions are less extreme than in "in-the-wild" videos, the practicality of the method could be severely limited if such cases are not handled well.

* It is also noticeable that lighting conditions and reflections on the glasses are transferred very directly from the reference image and do not account for the target scene’s surroundings, lighting, or color.

* Artifacts and temporal inconsistencies remain; the resulting videos are not always visually pleasant. For example, the glasses' temples flicker and the optical flow adjustment for eyewear pose lags behind head movements. While the approach makes significant progress, further improvements (e.g., increasing optical flow guidance weights or using more denoising steps) may be necessary to balance computational demands and the responsiveness of the real-world demo.

* The "mask-free" claim is questionable. The method may not require full facial masks for both the source and target, but it still requires a mask for the eyewear itself, so it cannot be considered completely mask-free.

**Questions:**

### Suggestions and Questions

* Could you provide videos of your model applied to faster, more chaotically moving subjects with a wider range of motion?

* Please also test transferring eyewear from a frontally lit face to a backlit one, or otherwise change the surroundings reflected in sunglasses. Consider proposing a metric sensitive to inconsistencies between the reflections in the target image and those generated by your method.

* If possible, try to apply your algorithms to other wearables/facial features that are easy to segment with enough precision. Maybe this can improve the scope of the paper.

---

> ### Author Response · Authors · 2025-11-20
>
> We thank the reviewer for the detailed and constructive feedback, as well as the positive remarks on our method, benchmark, and experimental evaluation. We address the raised concerns below.
>
> **W1. & Q1. Evaluation of more extreme cases**
>
> Thank you for this valuable suggestion. We agree that more challenging cases would help clarify the method’s practical limits.  In our experiments, we have observed that the model produces plausible transfers even when the target face undergoes yaw rotations of approximately ±45°. We are preparing additional results highlighting challenging scenarios and will include these examples in the revision.
>
> **W2. & Q2. Lighting condition issues**
>
> **Lighting variations**
>
> Thank you for the helpful suggestion. Our evaluation benchmark already includes cases where the target video is recorded under lighting conditions that differ noticeably from the reference image, and in our tests, the method handles such illumination differences reasonably well. We agree that including more challenging scenarios would provide clearer insight into the method’s behavior. In the revision, we will add qualitative examples spanning a broader range of lighting conditions, including cases such as transferring eyewear from a frontally lit reference to a backlit target video, as suggested.
>
> **Reflections in sunglasses**
>
> Regarding reflections, our current pipeline transfers the lens appearance directly from the reference image and does not attempt to estimate environment maps or target-scene illumination. Producing physically consistent reflections would require explicit modeling or learning of scene lighting, which is outside the scope of our training-free, reconstruction-based framework. We also note that existing image- and video-editing baselines do not synthesize environment-aware reflections either; most preserve the reference reflections or produce outputs that are not illumination-consistent (as shown in Fig. 3).
> We will clarify this limitation and add a brief discussion on potential future directions, including reflection-aware evaluation metrics. We view this as an interesting but orthogonal research direction that can complement our method in future work.
>
> **W3. Temporal inconsistency and artifacts**
>
> We appreciate the reviewer’s comment that the proposed approach makes significant progress toward practical eyeglass transfer. We fully agree that, for a polished real-world demo, additional engineering effort is needed to further suppress residual artifacts and to tune the trade-off between computational cost and output quality.
> Our primary focus in this work is on the algorithmic and scientific contributions, demonstrating that a training-free, target-mask-free, reference-guided framework built on DiffAE can achieve state-of-the-art temporal consistency and identity preservation for eyeglass transfer, accompanied by a dedicated benchmark and evaluation protocol.
> The system does include several tunable parameters, such as temporal attention settings and the number of denoising steps, that can be adjusted to further suppress minor artifacts, though at a higher inference cost. In the revision, we will add a brief discussion of this trade-off and explicitly frame these choices as engineering directions for practical deployment, rather than limitations of the proposed framework itself.
>
> **W4. “Mask-free” terminology**
>
> We agree that the original phrasing was ambiguous. We will revise it to “target-mask-free” to emphasize that no mask is required for the target video. Obtaining a per-frame target mask is extremely impractical for eyeglass transfer in video. Since the eyeglasses are *absent* in the target sequence, their precise position, shape, and occlusion relationships cannot be reliably annotated or inferred across many frames. This makes target-mask requirements unsuitable for real-world applications.
> The reference mask for eyeglasses is automatically obtained via Grounded-SAM and does not require manual annotation. We will update the wording accordingly.
>
> **Q3. Applying to other facial features/wearables**
>
> Thank you for the suggestion. Since the method operates through localized latent blending, it extends naturally to several face-centric attributes. We have preliminary results for additional facial features and will include examples as part of the revision.

---

> > ### Comment · Reviewer_f7Lq · 2025-11-28
> > **Comment on the authors response**
> >
> > Thank you for your response, while you assure that most of the mentioned issues are to be addressed in future revisions of the paper and upon publication of paper’s code and data, I decided to keep my score as it is stated, because such revisions of the paper might require resubmission. Keep up the good work, your findings are interesting and promising nonetheless.

---

> > > ### Author Response · Authors · 2025-11-28
> > > **Thanks! We are working on the revision right now.**
> > >
> > > Thanks a lot for the comments!
> > >
> > > We also appreciate your understanding regarding our response.
> > >
> > > > such revisions of the paper might require resubmission
> > >
> > > Since ICLR allows authors to submit revised papers, we are actually working on our revision.
> > > We expect to upload the new version, reflecting the above responses and additional experiments, within the next day or two.
> > >
> > > We would appreciate it if you could assess the new manuscript after we complete the work.
> > > We will notify the reviewers via comments as soon as our revision is ready :)
> > >
> > > Thanks again,
> > >
> > > Authors

---

> > > ### Author Response · Authors · 2025-12-02
> > > **Summary of Revisions in Response to Reviewer f7Lq**
> > >
> > > Dear Reviewer,
> > >
> > > Thank you again for your thoughtful review and for the encouraging comments in your discussion post. We truly appreciate your constructive feedback, which has helped us strengthen the paper.
> > > We would like to inform you that we have now uploaded a revised version of the manuscript. In this revision, we have carefully addressed all of your raised concerns, including:
> > >
> > > * **Extreme cases & robustness** (W1 & Q1)
> > >      We added new evaluations covering large yaw changes (~±40–50°), occlusions (hair, hand, object), and significant reference–target viewpoint mismatch (Sec 4.2, Appendix C.6). We have also included two new examples of eyeglass transfer to the target with more chaotic motion in our supplementary video.
> > >
> > > * **Lighting condition issues** (W2 & Q2)
> > >
> > >     We added an evaluation on transferring an indoor, front-lit reference to challenging lighting conditions, including direct sunlight, backlit, and side-lit targets (Sec 4.2). We also clarified the limitation of reflection transfer in training-free frameworks and added a discussion of future directions (Appendix G).
> > >
> > > * **Temporal consistency & artifacts** (W3)
> > >
> > >     We added a short note in Appendix B explaining the effect of tunable factors (e.g., temporal attention settings and denoising steps) and clarifying the trade-off between computational cost and artifact suppression in practical deployment.
> > >
> > > * **“Mask-free” terminology** (W4)
> > >
> > >      We refined the terminology to target-mask-free throughout the paper.
> > >
> > > * **Generalization beyond eyeglasses (Q3)**
> > >
> > >     We added examples showing that our method can extend to other facial attributes, including eyebrows, nose, and moustache (Sec 4.2). We also included a simple sequential-editing example to further illustrate the method’s flexibility (Appendix C.7).
> > >
> > > All new results, figures, and methodological clarifications are included in the revised manuscript and supplementary material.
> > > We sincerely appreciate your constructive feedback and your positive remarks about the promise of the method. Your comments have materially improved the quality and clarity of our work.
> > > Thank you again for your time and for contributing to this review process.
> > >
> > > Best regards,
> > >
> > > Authors

---

### Author Response · Authors · 2025-12-02
**General Comment**

Dear PCs, SACs, ACs, and Reviewers,

We would like to provide a brief summary of the review landscape and our revision. We also provide a revised manuscript, taking all the reviewers' comments into account.

* **One reviewer assigned a strong score of 8 with high confidence (5)** expressed very positive remarks on the clarity of the paper, the soundness of the method, the strong results, and the comprehensive evaluation.
* **The reviewer with a score of 4** also provided positive assessments on several key aspects, including the clarity of the paper, the soundness of the methodology, the strong results, and the comprehensive evaluation. Their concerns primarily focused on clarifications of terminology, the lack of robustness evaluations, and the broader applicability beyond eyeglasses. During the discussion, **they stated that they were satisfied with our responses, conditional on revision**, and regarded our findings as *interesting and promising*. They noted that they kept the score only because the required updates were substantial. We believe that the score would be better if the discussion phase were to continue.
* **One of the reviewers gave a score of 2** based on a **factual misunderstanding** regarding the benchmark scope (i.e., assuming target videos were restricted to small head poses). Their remaining comments on baselines, novelty, and robustness were fully addressed in the revision.
* **The remaining reviewer with a score of 2** raised broader concerns about scope, novelty framing, and robustness, which we addressed comprehensively. Our revision includes extensive robustness experiments, generalization results, methodological clarifications, and expanded ablations that address their points.


We addressed all questions, weaknesses, and suggestions raised by the other reviewers. We have prepared a revised manuscript that incorporates these clarifications and includes new experiments addressing robustness (lighting variation, occlusions, pose change, mask inaccuracies, viewpoint mismatch), generalization beyond eyeglasses, strengthened ablations, and improved methodological explanations. A high-level summary is provided below, and detailed point-by-point responses can be found in our replies to each reviewer.

---

> ### Author Response · Authors · 2025-12-02
> **Summary of Revision Updates**
>
> To address all the valuable comments and concerns of the reviewers, we have accordingly updated our manuscript and supplementary materials as follows. We highlighted our main updates in blue in our revision.
>
> ### Key Updates:
>
> 1. **Terminology correction.** [Reviewer f7Lq W4, r74s W1, NpfN W1]
>
>     Revised “mask-free” → “target-mask-free” throughout (title, intro, evaluation)
> 2. **Strengthened introduction.**  [Reviewer r74s W4, NpfN W2]
>
>     Revised the introduction to highlight our key empirical observation that DiffAE’s semantic latent space naturally supports geometric alignment.
>
> 3. **Clarification of mask usage.** [Reviewer r74s W1, xwtJ Q3]
>
>     Sec. 3.2: Stated that the reference mask is bilinearly downsampled to latent resolution.
>
>     Sec. 3.3: Clarified target ROI extraction from the aligned reference mask.
>
> 4. **Strengthened methodology.**
>
>     Sec. 3.2 [Reviewer r74s W3, xwtJ Q4]: Expanded motivation for stochastic latent blending, explaining its role in stabilizing transitions and reducing artifacts.
>
>     Sec. 3.3 [Reviewer xwtJ W2 & Q2]: Refined explanation of regional flow-guided SA and acknowledged prior related ideas.
>
> 5. **Clarification of evaluation protocol (Sec 4.1)** [Reviewer NpfN W3]
>
>     Stated clearly that target videos are unconstrained in pose, expression, and lighting; restructured the paragraph to remove ambiguity.
>
> 6. **Generalization experiment to other facial attributes. (Sec 4.2)** [Reviewer f7Lq Q3, r74s W2 & Q2, xwtJ W1 & Q1]
>
>     Added results on eyebrows, nose, and moustache, demonstrating the method’s flexibility beyond eyeglasses.
>
> 7. **New robustness experiments.**
>
>    We have newly included six robustness experiments to address the reviewers’ concerns.
>
>     * **Robustness to lighting variations. (Sec 4.2)** [Reviewer f7Lq W2 & Q2, r74s Q1]
>
>         A figure for transferring an indoor front-lit reference eyeglasses to various challenging lighting conditions, including direct outdoor sunlight, back-lit and side-lit.
>
>     *  **Occlusion handling (Sec 4.2)** [Reviewer f7Lq W1 & Q1, r74s Q1]
>
>        Provide the result of our method handling three representative occlusions, including hair bangs, hand motion covering the face, and object occlusions.
>
>    * **Robustness to pose changes. (Sec 4.2)** [Reviewer f7Lq W1 & Q1]
>
>         A new yaw-variation figure confirms that our method remains stable up to approximately ±40–50°
>
>     * **Robustness to mask inaccuracies. (Appendix C.5)** [Reviewer r74s Q3, NpfN Q2]
>
>         A new figure demonstrates stability under perturbed reference masks, including dilated masks, shifted masks, and rough hand-drawn masks.
>
>     * **Robustness to reference–target pose mismatch (Appendix C.6)** [Reviewer r74s Q4]
>
>         Added a figure showing that the method still produces plausible transfer even under large viewpoint differences (e.g., left-facing reference vs. right-facing target, significant pitch mismatch)
>
>     * **Sequential attribute editing (Appendix C.7)** [Reviewer r74s Q2]
>
>         A new figure demonstrates that our method supports multi-step editing by transferring a reference nose, followed by eyeglasses.
>
> 8. **Strengthened ablation. (Sec 4.2 Ablation Study)** [Reviewer r74s W4, W5]
>
>     * Added detailed discussion between metric behavior and qualitative results.
>     * Added ablations for regional per-frame SA and Regional STSA (Table 4) and show that our regional flow-guided self-attention improves preservation of eyeglass context and temporal consistency.
> 9. **Comparison with original DiffAE classifier-guided editing. (Appendix C.2)** [Reviewer r74s Q5]
>
>     Added evaluation of DiffAE classifier-guided editing and confirmed its limitations in reference fidelity and temporal consistency.
>
> ----
>
> ### Minor Revisions:
>
> 1. Added two faster-motion examples to the supplementary video.
> 2. Added implementation notes discussing trade-offs between artifact suppression and inference cost (Appendix B).
> 3. Added a discussion on reflection handling and future extensions (Appendix G).
> 4. Moved the temporal consistency figure + discussion from the main text to the Appendix.
> 5. Polished wording, captions, and layout throughout.
>
>
>
>
> We hope these revisions clearly address the reviewers’ concerns and improve the overall clarity and completeness of the paper.

---

> > ### Author Response · Authors · 2025-12-02
> > **Summary of Review Issues and Our Clarifications**
> >
> > To assist the new AC, below we summarize the key issues raised in the reviews and the core points clarified in our rebuttal.
> >
> > 1. **Benchmark Scope Confusion** [NpfN W3]
> >
> >     Target videos are not restricted to ±15°; they are **fully unconstrained**. Only reference images are filtered. We update Sec. 4.1 accordingly.
> >
> > 2. **Appropriateness of Baselines** [NpfN Q1]
> >
> >     OmniTry is the only eyeglass-specific baseline. Other baselines include localized object-editing methods suitable for this task, and clothing try-on systems are not directly comparable. Our method is training-free while most baselines require task-specific training.
> >
> > 3. **Scope of the Method** [f7Lq Q3, r74s W2 & Q2, xwtJ W1 & Q1]
> >
> >     Eyeglasses form a uniquely challenging and representative test case as they occlude identity-critical regions and require precise geometric alignment. The method itself, however, is not eyeglass-specific and naturally extends to other face-centric attributes. New examples (eyebrows, nose, moustache, and sequential edits) are added in Sec. 4.2 and Appendix C.7.
> >
> > 4. **Novelty & Scientific Contribution** [r74s W4, NpfN W2]
> >
> >     Our core novelty lies in the scientific discovery that DiffAE’s semantic latent space performs geometric and positional alignment without extra pose modeling, keypoints, or target masks, which enables a new paradigm of reference-guided, target-mask-free video editing. We distinguish this clearly from prior DiffAE editing and inpainting pipelines.
> >
> > 5. **Mask Usage & “Mask-Free” Terminology** [f7Lq W4, r74s W1, NpfN W1, xwtJ Q3]
> >
> >     The method is target-mask-free, utilizing only a single automatic reference mask. Target masks are impractical for eyeglass transfer due to the absence of objects. We clarified the terminology throughout the paper and added details of mask usage in Sec. 3.
> >
> > 6. **Motivation of Stochastic Latent Blending** [r74s W3, xwtJ Q4]
> >
> >     Blending only semantic latents introduces local noise and boundary artifacts, and stochastic latent blending stabilizes transitions between target and reference features and reduces artifacts; Sec. 3.2 is rewritten accordingly.
> >
> > 7. **Role of Regional Flow-Guided Self-Attention** [r74s W4, W5, xwtJ W2 & Q2]
> >
> >     Our method restricts flow-guided SA to the ROI to avoid global interference. Additional ablations validate this design. We also added citations to related methods.
> >
> > 8. **Robustness (Lighting, Occlusion, Pose, Mask Errors, Viewpoint)** [f7Lq W1 & Q1, W2 & Q2, r74s Q1, 3, 4, NpfN Q2]
> >
> >     New figures and results confirm robustness across various conditions, including lighting, occlusions, ±40–50° yaw, mask errors, and large ref–target viewpoint mismatches.
> >
> > 9. **DiffAE + Classifier Guidance** [r74s Q5]
> >
> >     Classifier-guided editing operates only at the attribute level. We added experiments that show it does not preserve reference eyeglasses or temporal consistency.
> >
> > 10. **Temporal Consistency & Artifacts** [f7Lq W3, r74s W6]
> >
> >     Some inconsistencies arise from the limitations of optical flow and the nature of training-free pipelines. Despite this, our method achieves the best realism and leads to the best temporal consistency among all baselines. Additional discussion added in the Appendix.
> >
> > We sincerely appreciate the efforts of the reviewers and the AC. Your insightful feedback has significantly strengthened the paper. We have addressed all concerns carefully in our revised submission, and we hope that this consolidated summary assists the AC in their evaluation. We offer our sincere respect and appreciation to everyone involved.

---

### Meta-Review · Area_Chair_BwkG · 2026-01-04

**Summary:**

First, the reviewers are concerned that the "mask-free" claim is inaccurate, as the method still requires a segmentation mask for the reference eyeglasses [f7Lq, r74s, NpfN]. Second, reviewers find the narrow scope and limited evaluation. Specifically, the method is evaluated on their own filtered benchmark, has small head pose changes, and missing challenging settings, including rotations, or fast motions [f7Lq, r74s, NpfN, xwtJ]. Reviewers also mentioned limited novelty, viewing the method as an incremental combination of existing techniques [r74s, NpfN, xwtJ]. Finally, the results still exhibit artifacts and temporal issues [f7Lq, r74s]. The AC also shares the concerns of reviewer xwtJ that the “application seems extremely small" compared to the sophisticated nature of the method. ” The proposed approach seems general and not limited to just eyeglasses.

**Reviewer Concerns:**

There are some clarifying questions that the rebuttal addressed, e.g., in the evaluation setup, some details in terminology. However, the key concerns of “mask-free” are not addressed, changing the claim to “target-mask-free” limits the contribution. Additionally, the authors respond to many of the concerns in “future direction, e.g., lighting conditions. The AC regards these concerns as outstanding, as deferring to the future does not resolve the issue. In the rebuttal, the authors claimed that the framework is object-agnostic and can handle different facial features or accessories, such as sunglasses, masks, eyebrows, or beards, simply by providing the corresponding reference mask. The AC believes it would be better to showcase these with a fair comparison with existing works.

**Reviewer Scores:**

Based on the rebuttal, the AC does not believe the reviewers will change their score if they were to participate fully in the discussion. Specifically, the key concerns regarding novelty and scope were not convincingly addressed. E.g., when the scope is narrow, clarifying the scope is narrow would not broaden the scope. Furthermore, the AC also agrees with the reviewers that the evaluation is limited.

---

### Decision · Program_Chairs · 2026-01-26

Reject